# Radiation-Induced Esophageal Cancer: Investigating the Pathogenesis, Management, and Prognosis

**DOI:** 10.3390/medicina58070949

**Published:** 2022-07-18

**Authors:** Athanasios Syllaios, Michail Vailas, Maria Tolia, Nikolaos Charalampakis, Konstantinos Vlachos, Emmanouil I. Kapetanakis, Periklis I. Tomos, Dimitrios Schizas

**Affiliations:** 1First Department of Surgery, Laikon General Hospital, National and Kapodistrian University of Athens, 11527 Athens, Greece; nh_reas@hotmail.com (A.S.); mike_vailas@yahoo.com (M.V.); schizasad@gmail.com (D.S.); 2Department of Radiotherapy, School of Medicine, University of Crete, 71110 Heraklion, Greece; mariatolia1@gmail.com; 3Department of Medical Oncology, Metaxa Cancer Hospital, 18537 Athens, Greece; nick301178@yahoo.com; 4Department of Surgery, University Hospital of Ioannina, 45500 Ioannina, Greece; vlachoskonstantinos@yahoo.gr; 5Department of Thoracic Surgery, Attikon University Hospital, National and Kapodistrian University of Athens, 12462 Athens, Greece; periklistomos@hotmail.com

**Keywords:** esophageal cancer, radiation, pathogenesis, treatment, prognosis

## Abstract

One of the most serious late side effects of irradiation is the promotion of tumorigenesis. Radiation-induced esophageal cancer (RIEC) can arise in a previously irradiated field, mostly in patients previously irradiated for thoracic malignancies such as breast cancer, Hodgkin and non-Hodgkin lymphomas, head and neck cancers, lung cancer, or previous esophageal cancer. RIEC is rare and accounts for less than 1% of all carcinomas of the esophagus. There are little data available in the current literature regarding pathogenesis, diagnosis, treatment, and outcome of esophageal cancer developed in a previously irradiated field. RIEC seems to represent a biologically aggressive disease with a poor prognosis. Although it is difficult to perform radical surgery on a previously irradiated field, R0 resection remains the mainstay of treatment. The use of neoadjuvant and adjuvant chemoradiotherapy remains very helpful in RIEC, similarly to conventional esophageal cancer protocols. The aim of this article is to elucidate this rare but challenging entity.

## 1. Introduction

Incorporation of combined modality treatment (including neoadjuvant chemotherapy, radiotherapy, and surgical resection) has been employed for the management of locally advanced esophageal and gastroesophageal junction cancers [1]. The trimodal approach can potentially achieve higher rates of (a) tumor shrinkage-downstaging, (b) negative tumor resection (R0) margins, and (c) pathological complete response (pCR), thus offering lower disease recurrence rate and overall survival improvement [2,3,4,5,6,7]. The combination of (a) imaging with targeted irradiation techniques (i.e., Intensity-modulated radiation therapy, volumetric modulated arc therapy, protons), (b) 4D-Computed Tomography (CT) planning or other respiratory motion management methods, and (c) biological targeting (i.e., Positron Emission Tomography (PET)/CT implementation in radiotherapy planning) can offer more accurate planning to target volume delineation margins that minimize healthy surrounding tissue dose (i.e., heart substructures and lungs) and limit early and late toxicity incidences [1,8,9,10,11,12]. As one of the most serious irradiation late side effects, the promotion of tumorigenesis should be considered in the optimal treatment decision-making of long-term cancer survivors that are at increased risk of developing second malignancies [3,13].

Radiation-induced esophageal cancer (RIEC) can arise in a previously irradiated area [3] and is considered a late radiation effect when it fits certain predetermined criteria, including the timing of RIEC development (>5 years after radiation), age at irradiation, dose received and volume of the irradiated area, type of irradiated organ, the origin from tissues within the irradiated field, radiation technique, the different histopathological features compared to the primary tumor, and individual and family history of cancer [3,13]. Those criteria seem to resemble Cahan’s criteria. Cahan et al. reported 11 cases of bone sarcomas arising in previously irradiated bones. The cases selected in their study fulfilled the following prerequisites: (1) microscopic or roentgenographic evidence of the nonmalignant nature of the initial bone condition, (2) the sarcoma must have arisen in the area included within the previously radiotherapeutic beam, (3) a relatively long (>5 years), asymptomatic latent period must have elapsed after irradiation and before the clinical appearance of the bone sarcoma, and (4) all sarcomas must have been proved histologically [14].

There are little data available in the current literature regarding pathogenesis, diagnosis, treatment, and outcome of esophageal cancer developed in a previously irradiated field. The aim of this narrative review is to investigate and elucidate this rare, albeit challenging, entity.

## 2. Epidemiology and Pathogenesis of RIEC

RIEC is rare and accounts for less than 1% of all carcinomas of the esophagus [15]. The first human case, reported by Slaughter in 1957, developed 27 years after the patient’s radiation exposure [16]. RIEC incidence increases mostly in patients previously irradiated for thoracic malignancies such as breast, Hodgkin (HL) and non-Hodgkin lymphomas (NHL), head and neck, lung, or previous esophageal cancer.

Females seem to be predisposed to RIEC compared to males. Each Gray (Gy) of irradiation increases the solid cancer rate by approximately 58% in females (90% CI, 43–69%, vs. 35% (90% CI, 28–43%) in males [13]. The median presentation age is 65 years, with poor smoking or alcohol intake history. RIEC development risk increases after a latent 10-year period after irradiation exposure [12] and becomes greater if radiation was performed during childhood, suggesting higher radiation carcinogenicity when the organ is developing [12,15].

The most frequent histological subtype that can be identified is constituted by moderately or poorly differentiated esophageal squamous cell carcinoma (ESCC) [12,15]. Nevertheless, adenocarcinoma accounts for approximately 16.2% of cases [12]. Markar et al. reported in their study more adenocarcinomas in the RIEC group in comparison to the non-RIEC group, probably due to differences in underlying etiology [12]. Most patients in the non-RIEC group had a history of ESCC related to tobacco consumption, exposing their upper airways, lungs, and esophagus to a second SCC through the ‘cancerization field’ concept [17].

A well-documented example of the carcinogenic effect of ionizing radiation is the Hiroshima and Nagasaki atomic bomb survivors [18]. However, the process of radio-carcinogenesis in secondary esophageal cancer is not clearly understood, and accurate risk models or pathogenic pathways do not exist.

Different degrees of acute esophagitis is the most important early side effect of radiotherapy [18], while fibrosis and scarred esophageal strictures are some of its late effects [19]. Irradiation-induced cell death mostly affects cells with rapid cell turnover [20]. During irradiation, the regeneration of the mucosal surface of the esophagus is impaired, leading to esophagitis and alterations including nuclear hyperchromasia or denudation, focal basal epithelial cell necrosis, epithelial swelling and erosions or edema, and petechiae in the lamina propria and in the submucosa. On the other hand, late side effects of radiotherapy include submucosal fibrosis, thickening of the wall of the esophagus, and chronic esophageal ulcers due to vascular insufficiency, resulting in motility disorders and stenosis/strictures/fistula, which are considered well-described radiation effects [20,21].

Boldrin et al. investigated the loss of heterozygosity and microsatellite instability (MSI) in RIEC compared to primary EC. A statistically higher loss of heterozygosity frequency at the 17q21.31 chromosome was found, highlighting the fact that radiation may result in DNA mutations [22]. Radiotherapy causes single and double-strand DNA breaks. Single strand breaks may be converted to double-strand breaks during cell replication. Double-strand breaks can lead to gene mutation and malignant transformation of the irradiated tissue cells [13]. Consideration must be given to concurrent use of some chemotherapeutic agents (i.e., platinum-based) due to the higher RIEC incidence through mucosal cells’ damage. This compound acts mainly through the creation of intra-strand crosslinks in the DNA, and cisplatin itself is considered to be an effective mutagen and carcinogen in vitro [23].

The intensity-modulated radiotherapy technique (IMRT) may be associated with greater rates of irradiation-induced secondary malignancies risk, mainly due to the higher amount of normal tissue exposure to low radiation doses of radiation [13]. The Image Guided Radiotherapy technique (IGRT), in order to verify patient set-up, may contribute to approximately 5–20% of the total dose to the surrounding normal tissues [13].

In a multicenter nested case-control study, Morton et al. demonstrated that the odds of developing ESCC were 780% greater in breast cancer survivors who received ≥35 Gy of radiation when compared to those who had not received any radiation [24]. In a retrospective cohort study of 220,806 breast cancer survivors, Ahsan et al. demonstrated a 305% increase in RIEC risk in patients who previously received radiation when comparing different follow-up time periods of <5 years and ≥15 years [25]. In the study conducted by Roychoudhuri R et al., the relative risk of RIEC development increased significantly >15 years after radiotherapy for breast cancer [26].

Nobel et al. reported 69 patients with radiation-associated ESCC. In their cohort, the overall mean time for the development of ESCC was 18.2 years. Patients with head and neck cancers received the highest median radiation dose (66 Gy) and had the shortest median time to development of cancer (10.7 years) [4]. Markar et al. reported 75 patients with RIEC. The median dose of external radiotherapy received on the tumor site was 45 (25–66) Gy, and the median interval between the primary cancer and the esophageal cancer occurrence was 13 (6–39) years [12].

In a study conducted by Micke et al., several cases of RIEC after laryngeal carcinoma, breast cancer, esophageal cancer, lymphosarcoma, thyreotoxicosis, bronchial carcinoma, mediastinal neoplasms, HL, and NHL were reviewed. They included 66 patients in their study who received doses between 18.6 and 68 Gy (median dose 40 Gy) as primary treatment. The interval between radiation exposure and occurrence of the secondary esophageal carcinoma was a median of 15 years (ranging from 2 to 63 years). Most cases were moderately or poorly differentiated SCCs [15].

In another study conducted by Boldrin et al., a total of 20 metachronous esophageal cancers that developed after radiation for HL or breast cancer were reported. Metachronous esophageal cancers arose after a median period of 20 years (range 3–40). A high RIEC prevalence was observed in women with a past history of HL, suggesting a link between the occurrence of a metachronous esophageal carcinoma and previous therapy. Sporadic esophageal cancer usually had a lower incidence in women than in men; however, the pathogenesis of the aforementioned association is still unknown [22].

Scholl et al. reported 5 women who had primary esophageal cancer after having breast cancer radiotherapy. Radiotherapy was given over 26 to 65 days (mean 45.4) with total doses of 36 to 60 Gy (mean 50.6 Gy) in 18 to 25 fractions (mean 20.2). The time between radiotherapy and esophageal cancer varied from 13 to 31 years. Esophageal cancers developed at sites within or near radiation fields (also due to scatter radiation exposure) [27]. Eleven cases of cancer of the thoracic esophagus developing after postoperative radiation therapy for breast cancer were also reported by Ueda et al. The administered dose ranged from 35 Gy to 60 Gy, and the dose received by the thoracic esophagus was estimated between 10 Gy and 48 Gy. All cancer sites were involved in the irradiation field. The latent intervals from radiation to the manifestation of cancer ranged between 10 and 19 years [28].

Knowing whether the time of ESCC development is affected by radiation dose or patient age may help towards more effective long-term surveillance programs for previously irradiated patients [2]. Table 1 demonstrates the studies exhibiting the total radiation dose and time interval between irradiation and RIEC development.

## 3. Management of RIEC

Several studies have tried to elucidate the optimal therapeutic approach for this disease to guide the clinical management of these patients better. A Multidisciplinary Tumor Board of a high-volume EC center has a crucial role in optimal therapeutic management. Clinical staging using a combination of upper gastrointestinal endoscopy, Endoscopic Ultrasound (EUS), Positron Emission Tomography-Computed Tomography (PET/CT), and Computed Tomography (CT) scan are useful tools in determining the consistent variations in terms of diagnosis, assessment of resectability, and response to treatment.

The cornerstone of treatment in RIEC is complete surgical resection in a non-metastatic setting. If a radical surgical approach is not possible because of tumor extension, radiation toxicity (i.e., severe fibrosis), or poor patient performance status, radical re-irradiation or palliative chemotherapy should be considered. Radiation-induced malignancies are not less sensitive to chemotherapy and/or radiotherapy. External beam re-irradiation may not be technically feasible, mainly due to overlapping with the initial fields and the presence of high spot areas. However, endoluminal brachytherapy can provide a safe alternative [15].

There is a lot of skepticism concerning the effectiveness of neoadjuvant chemoradiotherapy in esophageal cancer developed in an irradiated field. Thus, a significantly reduced utilization of neoadjuvant chemotherapy (19.1% versus 47%; *p* < 0.005) has been reported in the post-radiation EC group [12]. Moreover, a three-stage procedure is commonly used in these patients, due to tumor location in the upper-third of the esophagus [12]. A greater incidence of R1/2 margins (21.3% versus 10.9%; *p* < 0.001), with a vertical margin more commonly involved (11.0% versus 4.3%; *p* = 0.001) in post-radiation EC patients, but no significant difference in lateral margin involvement between the two groups has also been reported. Increased in-hospital (14.0% versus 7.1%; *p* = 0.003) and 90 days (14.0% versus 6.9%; *p* = 0.002) mortality, overall morbidity (68.4% versus 56.4%; *p* = 0.006), surgical site infection (22.8% versus 14.5%; *p* = 0.009), neurological complications (17.6% versus 4.7%; *p* < 0.001), and an increased median length of hospital stay (29.3 days versus 24.9 days; *p* = 0.019) have also been reported in the first group [12].

In a study conducted by Nobel et al., patients with RIEC presented with earlier stage disease than patients with primary ESCC (stage II disease 50.7% vs. 28.5%, *p* = 0.002) [4]. On the other hand, more patients with primary ESCC presented with stage III disease (63.2% vs. 44%, *p* = 0.01). There was no difference in tumor location or grade. RIEC patients received surgery alone more frequently (20% vs. 7.3%) and definitive chemoradiation less often (52% vs. 65.9%) than patients with primary ESCC (*p* = 0.012). Additionally, RIEC patients were less likely to receive any neoadjuvant therapy (58.3% vs. 78.6%, *p* = 0.039) and moreover, were much less likely to receive specifically neoadjuvant chemoradiation therapy (29.2% vs. 72.9%, *p* < 0.001) [4]. Among patients receiving neoadjuvant therapy in the same study, there is a higher frequency of pCR in patients with primary ESCC compared to RIEC ones (38.5% vs. 16.7%, respectively; *p* = 0.024), with no difference in nodal positivity, lymphovascular invasion, or margin status. However, RIEC patients may have a higher rate of neural invasion (61% vs. 25.4%, *p* = 0.001). These results possibly implicate different tumor characteristics and cancer biology between RIEC and primary ESCCs [4]. The authors concluded that patients with previous malignancies are more likely to be in frequent contact with healthcare providers and be enrolled in regular follow-up, explaining why RIEC is identified in earlier stages than primary EC cases in this cohort [4].

In a retrospective study by Micke et al., 66 RIEC patients were reported. Most patients received surgical resection, 14 patients underwent radiotherapy alone, 2 had chemotherapy only, 1 patient received hyperthermic chemotherapy, and 1 patient received definitive high-dose-rate brachytherapy with Iridium-192 [15]. A total of 42.3% of the surgically treated patients achieved long-term survival, while most patients without surgery died within a few months. Although patients managed with chemoradiotherapy or brachytherapy did not achieve long-term survival, some patients achieved long-term remissions of cancer. The authors suggested that when a radical surgical approach is not feasible (i.e., in patients with large tumor extension or poor performance status), re-irradiation alone or palliative chemotherapy can be carried out as an alternative palliative method [15].

Ueda et al. reported that 11 RIEC thoracic esophagus patients developed after postoperative radiation therapy for breast cancer. In 8 patients, esophagectomy was performed, while 2 patients were managed by irradiation only. Additionally, 4 of the surgically treated patients received postoperative radiation, achieving long-term survival of 13 years, even with aggressive tumor characteristics. Patients managed with re-irradiation alone achieved survival of 13 months. Re-irradiation treatment may offer an alternative palliative therapy when surgery is not feasible. The authors highlighted the fact that radiation therapy is beneficial to radiation-induced cancers of the thoracic esophagus [28].

Finally, in a study by Taal., 8 patients with RIEC were re-irradiated. Not all patients were eligible for full dose radiotherapy, and the reported survival was between 2 and 13 months, concluding that none of the radiation-induced EC patients could be cured, but in the absence of a radical surgery, re-irradiation may be an option [29]

## 4. Prognosis

As far as prognosis is concerned, lower 5-year overall (28.8% vs. 50.5%, *p* = 0.003), event-free (32.2% vs. 42.5%, *p* = 0.002), and cancer specific survival (46.1% vs. 52%, *p* = 0.027) has been reported in the radiation-induced EC groups compared to patients with primary EC. This can possibly be explained by the rare use of neoadjuvant chemoradiotherapy in these patients [12]. However, when R0 resection is achieved, there seems to be no significant difference in overall (31.6% vs. 40.5%, *p* = 0.937), locoregional (12.3% vs. 22.9%, *p* = 0.967), and distant (18.8% vs. 11.8%, *p* = 0.263) tumor recurrence. Also, subset analysis conducted including only patients adherent to Cahan’s criteria led to the detection of comparable oncological outcomes (OS, DFS) between RIEC and primary EC patients [12].

In another study comparing radiation-induced EC patients (Group A) to patients with primary EC (Group B), the authors did not report lower overall survival for the radiation-induced group [4]. However, they reported that among patients who were treated with surgery alone, those in Group A had lower overall survival (5-year OS of 15% vs. 33%; *p* = 0.045) and higher risk of recurrence (55% vs. 25%) compared to those in Group B. These facts indicate that radiation-induced EC is probably a more aggressive disease [4]. The authors also performed a subgroup analysis of different treatment strategies in Group A patients. Group A patients treated with surgery alone had lower OS compared to group A patients receiving neoadjuvant therapy plus surgery [4]. Moreover, Group A patients treated with surgery alone or patients receiving definitive chemoradiation had a higher risk of recurrence (5-year recurrence risk of 55% and 45%, respectively) compared to Group A patients treated with neoadjuvant therapy plus surgery (15%). The overall survival and risk of recurrence between those treatment groups highlight the benefit of neoadjuvant therapy in radiation-induced EC patients receiving surgical management [4].

Finally, in a recent study published by Pierobon et al., a case-matched comparative study was contacted between RIEC and primary esophageal cancer patients. They concluded that RIEC and primary esophageal cancer patients showed comparable results in terms of exposure to neoadjuvant treatment, surgical radicality, long-term disease-free (36.1% vs. 47.3%; *p* = 0.39), and overall survival (30.7% vs. 35.7%; *p* = 0.44) in the two cancer groups. [30]

Table 2 presents a summary of the main therapeutic options and clinical outcomes reported in studies with RIEC patients.

## 5. Conclusions

In conclusion, RIEC seems to represent a biologically aggressive disease with an underlying poor prognosis. The often reported short survival is mainly related to reduced utilization of neoadjuvant chemoradiotherapy and increased incidence of tumor margin involvement in surgical specimens. Surgery remains the mainstay of treatment in this special population of RIEC patients, achieving satisfying results when R0 resection is performed, but due to potential clinical consequences for patients, the decision should be based on a dynamic tug-of-war between complications and survival benefit in a Multidisciplinary Tumor Board. Neoadjuvant chemotherapy or chemoradiation should be used when possible. Multicenter clinical studies are also important in establishing the optimal treatment algorithm to offer these patients long-term survival.

## Figures and Tables

**Table 1 medicina-58-00949-t001:** Total radiation dose and time interval between irradiation and development of radiation-induced EC.

Author	Year	Radiation Dose	Time Interval
Nobel et al. [4]	2019	66 Gy	18.2 years
Markar et al. [12]	2017	45 (25–66) Gy	13 (6–39) years
Boldrin et al. [22]	2015	N/A	20 (3–40) years
Morton et al. [24]	2012	≥35 Gy	N/A
Roychoudhuri et al. [26]	2004	N/A	increased risk >15 years after radiotherapy for breast cancer
Scholl et al. [27]	2001	50.6 (36–60) Gy	13–31 years
Micke et al. [15]	1999	40 (18.6–68) Gy	15 (2–63) years
Ahsan et al. [25]	1998	N/A	305% increase in risk ≥15 years after radiotherapy
Ueda et al. [28]	1991	35–60 Gy	10–19 years

**Table 2 medicina-58-00949-t002:** Therapeutic options and main clinical outcomes in studies with post-radiation esophageal cancer patients.

Author	Year	Patients	Therapy	Main Clinical Outcome of Radiation-Induced EC Compared to Primary EC
Pierobon et al. [30]	2022	51	- comparable results to neoadjuvant treatment - comparable surgical radicality	- same OS, DFS - more postoperative complications (pulmonary)
Nobel et al. [4]	2019	69	- surgery alone more frequently - definitive chemoradiation less often - less likely to receive any neoadjuvant therapy	- same OS - higher risk of recurrence
Markar et al. [12]	2017	75	- reduced utilization of neoadjuvant chemotherapy - R0 resection	- greater incidence of R1/2 margins - increased morbidity, mortality, LOS - lower 5-year OS, DFS, LR
Micke et al. [15]	1999	66	- 49 patients surgery - 14 patients radiotherapy - 2 patients chemotherapy - 1 patient hyperthermic chemotherapy	- long-term OS, DFS with radical surgery - when radical surgery not possible, reirradiation alone or palliative chemotherapy can be carried out
Taal et al. [29]	1993	8	reirradiation	poor survival
Ueda et al. [28]	1991	11	- 8 patients esophagectomy +/− irradiation - 2 patients irradiation	- surgery+ reirradiation achieves long-term OS

## Data Availability

Not applicable.

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
