# Peer review of "Radiation-Induced Esophageal Cancer: Investigating the Pathogenesis, Management, and Prognosis"

_medicina, 2022, doi:10.3390/medicina58070949_

Round 1

Reviewer 1 Report

This paper addresses a very rare radiation-induced esophageal cancer.

The topic addressed is interesting and deserves a constructive discussion. The data presented in this paper may help physicians treating radiation induced esophageal cancer provide information to their patients. I think this paper deserves acceptance.

Author Response

We thank the reviewers for the valuable time and effort spent to review our manuscript. Their comments and suggestions definitely improved the quality of our article.

Reviewer 1:

This paper addresses a very rare radiation-induced esophageal cancer. The topic addressed is interesting and deserves a constructive discussion. The data presented in this paper may help physicians treating radiation induced esophageal cancer provide information to their patients. I think this paper deserves acceptance.

Response: We thank the reviewer for his positive comments concerning the data presented and the content of our article.

Reviewer 2 Report

Some revisions may significantly improve the quality of the paper:
1. When discussing the treatment options and the prognosis of RIEC, the authors, correctly, refer to the retrospective cohort of the Memorial Sloan Kettering Group (Nobel et al. JOGS 2019) and the multicentric study by the FREGAT working group (Markar et al. EJC 2017). Based on the results of these studies, the authors mention a higher postoperative morbidity, a reduced exposure to neoadjuvant treatment and, consequently, a reduced overall and cancer-specific survival in the RIEC groups compared to patients with primary esophageal cancer (EC). While the quality of the aforementioned studies is undisputable, more recent data from a single referral center, comparing the results of multimodal treatment for RIEC and primary EC by means of a case-matched analysis, report comparable exposure to multimodal treatment and a comparable long-term survival in the two cancer groups (please see Pierobon ES, Capovilla G et al. International Journal of Surgery 2022 - https://doi.org/10.1016/j.ijsu.2022.106268). The results of this study should be included in the discussion regarding the feasibility of multimodal treatment for RIEC and its prognostic implications.
2. Please refer to Cahan’s criteria when discussing the definition of RIEC. (W.G. Cahan, H.Q. Woodard, N.L. Higinbotham, F.W. Stewart, B.L. Coley, Sarcoma arising in irradiated bone, Cancer. 1 (1948) 3–29. https://doi.org/10.1002/(sici)1097-0142(19980101)82:1<8::aid-cncr3>3.0.co;2-w.). Also, in the study by Markar et al. (Markar et al. EJC 2017), the subset analysis conducted including only patients adherent to Cahan’s criteria led to the detection of comparable oncological outcomes between RIEC and primary EC. This should also be mentioned in the discussion about the prognostic relevance of RIEC.
3. Page 2 line 52: please mention that this is a “narrative“ review.
4. Page 4 line 143: the word “field” is misspelled.

Author Response

Authors’ response to the reviewers’ comments

We thank the reviewers for the valuable time and effort spent to review our manuscript. Their comments and suggestions definitely improved the quality of our article.

Reviewer 2:

Some revisions may significantly improve the quality of the paper:

  1. When discussing the treatment options and the prognosis of RIEC, the authors, correctly, refer to the retrospective cohort of the Memorial Sloan Kettering Group (Nobel et al. JOGS 2019) and the multicentric study by the FREGAT working group (Markar et al. EJC 2017). Based on the results of these studies, the authors mention a higher postoperative morbidity, a reduced exposure to neoadjuvant treatment and, consequently, a reduced overall and cancer-specific survival in the RIEC groups compared to patients with primary esophageal cancer (EC). While the quality of the aforementioned studies is undisputable, more recent data from a single referral center, comparing the results of multimodal treatment for RIEC and primary EC by means of a case-matched analysis, report comparable exposure to multimodal treatment and a comparable long-term survival in the two cancer groups (please see Pierobon ES, Capovilla G et al. International Journal of Surgery 2022 - https://doi.org/10.1016/j.ijsu.2022.106268). The results of this study should be included in the discussion regarding the feasibility of multimodal treatment for RIEC and its prognostic implications.

Response: We thank the reviewer for his useful comment. We have added the data from the study by Pierobon et al. at the end of ‘Prognosis’ section. The main clinical outcomes from their study have also been summarized at ‘Table 2’.

  1. Please refer to Cahan’s criteria when discussing the definition of RIEC. (W.G. Cahan, H.Q. Woodard, N.L. Higinbotham, F.W. Stewart, B.L. Coley, Sarcoma arising in irradiated bone, Cancer. 1 (1948) 3–29. https://doi.org/10.1002/(sici)1097-0142(19980101)82:1<8::aid-cncr3>3.0.co;2-w.). Also, in the study by Markar et al. (Markar et al. EJC 2017), the subset analysis conducted including only patients adherent to Cahan’s criteria led to the detection of comparable oncological outcomes between RIEC and primary EC. This should also be mentioned in the discussion about the prognostic relevance of RIEC.

Response: We thank the reviewer for his important comment. We refer in the revised manuscript to Cahan’s criteria in the ‘Introduction’ section when discussing the definition of RIEC. We have also mentioned the subset analysis of Markar’s et al. study in the relevant ‘Prognosis’ section.

  1. Page 2 line 52: please mention that this is a “narrative” review.

Response: We thank the reviewer for his comment. We have stated that this is a narrative review in the revised manuscript in the ‘Introduction’ section.

  1. Page 4 line 143: the word “field” is misspelled.

Response: We thank the reviewer for his comment. We have corrected the misspelled word in the revised manuscript.
